# Domain Bridging: Enabling Adaptation without Peeking at Target Data

## Abstract

Adapting models to target domains with proprietary data remains a challenging problem. One possible setup to enable adaptation is to allow target domain owners to privately evaluate candidate models on their own data. For example, model providers consider how to adjust models to better fit the unseen target data, relying solely on returned model performance. Existing methods adopt Zeroth-Order (ZO) optimization to refine model parameters or employ a two-stage learning process that first identifies the target-related samples in the source data and then retrains the model. However, we find that these methods struggle to generalize well for the target tasks during inference, primarily because of the failure to account for data-statistical shifts between source and target domains. To address this limitation, we introduce the concept of *domain bridging* in the context of model adaptation for proprietary target data. The core idea is to bridge the domain gap by learning target-aligned perturbations on source data, enabling the fine-tuned model to achieve better performance on target domains. A natural attempt is to extend ZO optimization to this setting. However, this approach fails to produce reliable perturbations on real datasets. To address this, we design a target-aligned, sample-wise perturbation learner, enabling reliable adaptation from performance-only feedback. We provide theoretical convergence guarantees and demonstrate through experiments on five datasets across image and text modalities that our domain bridging method achieves state-of-the-art performance, improving accuracy by approximately 4%.

## 1 Introduction

Adapting a pre-trained model from source data to downstream target data is an essential technique for improving model performance in diverse real-world scenarios (Pan & Yang, 2009; Kouw & Loog, 2018; Zhuang et al., 2020). By aligning models with the specific characteristics of a target domain (Ganin & Lempitsky, 2015), adaptation ensures robust performance in dynamic and application-specific environments.

While standard fine-tuning suffices when target data are accessible, real deployments often preclude such access. In classification tasks, a pre-trained model can be adapted via fine-tuning if features and labels from the target domain are available. However, this straightforward technique becomes challenging when target data is inaccessible (Li et al., 2023; Szép et al., 2024), particularly in privacy-sensitive domains such as healthcare, finance, and e-Commerce, where strict data privacy regulations prohibit the sharing of proprietary information. Healthcare providers, for instance, are barred from sharing patient records (Newman, 2008), while financial institutions face similar constraints when adapting fraud detection models (Găbudeanu et al., 2021).

Existing strategies to address this challenge fall into three broad categories. The first category relies on a trusted third-party environment (Melissourgos et al., 2022) where target data is essentially fed to the pre-trained model for back-propagation, including tuning model adapters in an offsite manner (Xiao et al., 2023). The second category leverages encryption techniques, such as homomorphic encryption (Fan & Vercauteren, 2012), which have been applied primarily to model inference (Chen et al., 2022; Liu & Liu, 2023). Applying these techniques to training or tuning remains computationally prohibitive, although a recent study has explored encryption-friendly tuning paradigms (Rho et al., 2024). The third is the evaluation-based setup (Li et al., 2023), where the target domain owner can assess candidate models on their local data and return aggregate feedback, such as accuracy or

Figure 1: **Domain Bridging** (DB): We study how perturbing the source data can facilitate fine-tuning the source model to eventually adapt to the unobserved target data. The key component here is designing an effective gradient estimation method. Note that in practical scenarios, only model parameters will be sent to clients' side to save the communication cost.

error rates, to model providers. In this work, we position our approach within the third category, formulating model adaptation as an optimization problem over performance feedback under privacy constraints.

A primary approach for evaluation-based model adaptation is Zeroth-Order (ZO) optimization (Spall, 1992; Wierstra et al., 2014), which searches the parameter space without gradients (Li et al., 2023; Malladi et al., 2023). Although it is source-free and simple to deploy, ZO often yields noisy or unstable search directions in high-dimensional spaces, resulting in poor target-domain generalization at inference (Zhang et al., 2024b). An alternative line incorporates source data into the adaptation process. Data Shapley (DS) (Ghorbani & Zou, 2019) scores source examples using a black-box evaluation oracle and selects task-relevant points for retraining the model, thereby improving alignment to the target domain. Although DS is not initially designed for adapting a pre-trained model to unseen target data, we recognize that this method can serve as a new baseline. However, decoupling data selection from adaptation can introduce stage misalignment and inefficiency, which propagates errors and ultimately limits inference-time performance.

To address these limitations, we propose the concept of *domain bridging*, a novel concept that leverages source data for privacy-preserving adaptation to proprietary target domains. Our approach focuses on how perturbing the source data can effectively bridge the gap between the source data and the unobserved target data – an aspect overlooked by existing methods. By introducing data perturbations, we establish a more reliable link between the model parameters and performance feedback from the target data. With the perturbed source data, we are able to fine-tune the pre-trained model as if we had access to the target data. It is important to note that the perturbed data are not intended to replicate the target data in appearance or distribution; rather, they are optimized to steer the model towards learning feature representations that align more closely with those of the unseen target data. This design preserves privacy, as no target examples or distributions are disclosed. By reframing the bi-level objective to prioritize representation alignment over direct parameter search, our method facilitates adaptation in proprietary-data settings.

The key contributions of this work are threefold.

- We introduce the concept of domain bridging, which seeks to narrow the source-target domain gap by learning source data perturbations guided by performance on unobserved target data, offering a novel approach to adaptation without direct access to target data.
- We propose an efficient domain perturbation generation method that solves a non-differentiable bi-level optimization problem through reliable gradient estimation, which closely couples parameter updates with returned target-side performance feedback. We also provide proof for convergence guarantees of our approach.
- We demonstrate the effectiveness of our method by comparing it with existing solutions on both image and text classification tasks. Our method achieves competitive or superior state-of-the-art results while requiring fewer target-side query resources.

It is worth noting that our work addresses the fundamentally different evaluation-based domain adaptation problem, where model adaptation relies solely on performance feedback from the target domain without accessing individual target samples. This critical distinction sets our approach apart from traditional domain adaptation methods. For a detailed comparison of our approach with existing domain adaptation methods and other approaches in adjacent areas, please refer to Appendix A.

## 2 PROBLEM STATEMENT

We focus on a model adaptation problem, where a classifier $f$ parameterized by $\theta$ is trained on source data $(X_S, Y_S) = \{x_i, y_i\}_{i=1}^n$ ($x_i \in \mathbb{R}^d$) with a loss function $\ell(\cdot)$ (e.g. cross-entropy loss), and we aim to adapt it to unobserved target data. Throughout this document, we focus on the case where the source and target domains share the same label space, thus the model architecture remains unchanged during adaptation. To enable the adaptation, the target data is partitioned into two subsets: support set and holdout set. The support set is utilized to evaluate the model performance during adaptation, denoted by $V(\theta)$, whereas the holdout set serves to assess the model's generalization capabilities. Without loss of generality, assume that smaller values of $V(\theta)$ indicate better performance. For simplicity, all gradients with respect to $\theta$ are presented without explicitly showing $\theta$. Our goal is to update $\theta$ to adapt to unobserved target data using limited feedback on model performance.

### 2.1 PERFORMANCE GUIDED SEARCH

An effective solution to this adaptation problem is to search for the optimal $\theta$ in the parameter space based on the evaluation performance, also referred to as Performance Guided Search (PGS) (Li et al., 2023). Due to the lack of information about the support set, the exact gradient of the evaluation function $V(\cdot)$ with respect to $\theta$ cannot be obtained. PGS adopted Zeroth-Order (ZO) optimization estimator (Duchi et al., 2015; Nesterov & Spokoiny, 2017) to compute the gradient, which is in a form of

$$\hat{\nabla} V(\theta) = \mathbb{E}_z \left[ \frac{V(\theta + \epsilon z) - V(\theta - \epsilon z)}{2\epsilon} z \right], \tag{1}$$

where $z \in \mathbb{R}^{|\theta|}$ is a random direction typically drawn from the standard Gaussian distribution $\mathcal{N}(\mathbf{0}, \mathbf{I})$, and $\epsilon$ serves as the perturbation scale. Note that $V(\theta \pm \epsilon z)$ in Eq. (1) refers to mirroring the unobserved evaluation process on support set, and expectation over $z$ can be achieved by Monte Carlo approximation, implying the query efficiency complexity in real scenarios. Although more samplings to the estimator better preserves the gradient length (Ilyas et al., 2018) and reduces variance (Duchi et al., 2015; Liu et al., 2018b), sampling once only for a single layer update in deep neural networks has been shown to be efficient (Malladi et al., 2023).

Compared to random searches (Bergstra & Bengio, 2012), the idea of PGS is to search for a more promising direction $\hat{\nabla} V(\theta)$ in the parameter space which can guide the source model to gradually adapt to the unobserved target data based on the scores of $V(\cdot)$. Recent works also extend PGS to one-to-many settings (Feng et al., 2023; Zelikman et al., 2023), where multiple proprietary target data are considered.

Updating model parameters through ZO optimization for adaptation can be feasible, but it has two key limitations. (1) In the context of deep neural networks, the layer-by-layer update strategy does not align with the actual gradient chain due to its backpropagation-free nature, leading to significant fluctuation and slow convergence. (2) Adjusting model parameters in an unbounded parameter space based solely on numerical evaluation scores may result in overfitting to the support set and generalizing poorly on the holdout set. We demonstrate these findings through experimental observations on Office-31 (A-W), as shown in Fig. 2. The result shows mean accuracy over 10 runs with standard deviation shaded.

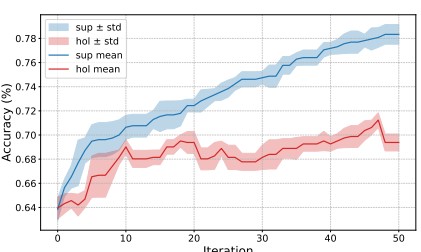

Figure 2: Accuracy curves of PGS on support set and holdout set, respectively.

### 2.2 RETRAINING WITH SOURCE DATA VALUATION

PGS can be loosely regarded as a source-free method if we overlook the fact that a few works (Yvinec et al., 2023; Zhang et al., 2024a) also utilized source data to identify the key layers for fine-tuning to achieve efficient adaptation. When source data are accessible, as is common in scenarios such as the "*model provider and client user*" setting, we recognize that the adaptation problem to unobserved target data can have other solutions.

Data Shapley (DS) (Ghorbani & Zou, 2019) equitably determines the value of each datum for the source data by quantifying the expected contribution to the model performance on a target set. As the model performance assessment is a black box oracle, this method can indicate high value source data on average through model performance $V(\theta)$. A scalable implementation for DS involves iterative updating of the source data values on a random permutation of them. For example, in the $t$-th iteration with a random permutation of all source data points $\pi^t$, the value of $i$-th point is updated through the following scheme:

$$\phi_{\pi^t[i]} = \frac{t-1}{t}\phi_{\pi^{t-1}[i]} + \frac{1}{t}(V(\theta_i^t) - V(\theta_{i-1}^t)), \tag{2}$$

where the second term on the right-hand side measures the marginal contribution of $i$-th point, and $\theta_i^t$ denotes that performance feedback of the model trained with the $i$-th point included. After convergence, these DS values are used to train a new classifier with a reweighted loss. We refer to this method as Retraining with Source Data Valuation (RSDV) in the following sections.

*Remark.* The key feature of RSDV is that the source data values are obtained during the optimization of the source model, which is slightly misaligned with our problem setting. A similar but earlier approach to acquire individual weights over a pre-trained model is the use of the influence function (Koh & Liang, 2017). However, DS derives equitable data values that are more reliable in reweighting individual loss based on the public code package[1].

We summarize that RSDV introduces the use of source data for model adaptation, enabling the training of a new model in a standard forward-backward manner on an adjusted source-data distribution. Please refer to its improved performance over PGS in Section 4. However, RSDV is a two-stage approach in which the data values depend on the original source tasks (that is, the use of $V(\theta_i^t)$). For example, considering that we have sufficient resources to re-evaluate the data values during training with the reweighted loss, we may obtain a new but different set of source data values, potentially indicating a less optimal adaptation performance of RSDV.

## 3 DOMAIN BRIDGING

By considering both the strengths and limitations of PGS and RSDV, we propose the concept of Domain Bridging (DB), shown in Fig. 1, by which we focus on how perturbing the source data can help bridge the gap between the source data and the unobserved target data. In other words, we anticipate that incorporating source data during adaptation will better capture potential domain shifts because, ideally, if we have unlimited chance to exploit what the target data looks like, we are able to achieve a successful adaptation eventually.

### 3.1 PERTURBATION WITH ZO OPTIMIZATION

Inspired by PGS, we can essentially estimate the gradient of $V(\cdot)$ w.r.t. the source features $X_S$ via ZO optimization. Following the estimator in Eq. (1), we can write

$$\frac{\partial V(\theta)}{\partial X_S} \approx \mathbb{E}_\Delta\left[\frac{V(\theta^*(X_S + \epsilon\Delta)) - V(\theta^*(X_S - \epsilon\Delta))}{2\epsilon}\Delta\right], \tag{3}$$

where $\Delta$ is a sampled random perturbation with the same size of $X_S$, $\epsilon$ is overloaded as the step-size scalar, and $\theta^*(X_S + \epsilon\Delta)$ represents the optimal classifier trained on the perturbed source data $(X_S + \epsilon\Delta, Y_S)$, and similar to $\theta^*(X_S - \epsilon\Delta)$. Let $\mathcal{X}$ be the feasible set of input features (e.g., image pixels are in the range of $[0, 255]$), and we will get perturbed features $X_S'$ by a projected gradient descent,

$$X_S' \leftarrow \text{Proj}_\mathcal{X}\left(X_S - \eta\frac{\partial V(\theta)}{\partial X_S}\right). \tag{4}$$

We then replace $X_S$ with $X_S'$ and repeat the process described above for multiple rounds, continuing until convergence or until the budget is exhausted. We perform a toy experiment for this method following the configuration of Li et al. (2023), where both the source and target data are generated from Gaussian distributions, but with different statistics. Initially, a three-layer perceptron model is trained on the source data, which fails to generalize to the target data, as presented in Fig. 3a.

---

[1]https://github.com/uvanlp/valda

Following Eq. (4), we add the learned perturbations to the source data progressively and update the model accordingly. After 80 times of evaluation on target data, the model achieves perfect adaptation, as shown in Fig. 3b. By visually inspecting the distribution shift, we observe that the source data move closer to the target data without directly access, validating the feasibility of our DB concept.

To scale up to real datasets, we use a mini-batch or a single data point (as is done in DS) of $X_S$ for a single-step update to approximate $\theta^*(\cdot)$. However, the adaptation performance is usually not satisfactory due to the less precision of gradient estimation of Eq. (3). We provide evidence of its poor ability to scale up to real datasets by monitoring its performance curves in Appendix D.1. Intuitively, from an optimization perspective, the process of obtaining $\theta^*(\cdot)$ involves a white-box model learning, i.e., through backpropagation, while the gradient information in Eq. (3) does not flow through $\theta$ and the computation of $V(\theta^*(X_S \pm \epsilon\Delta))$ is simply treated as a black-box process.

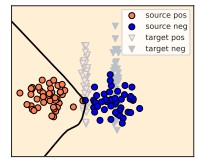 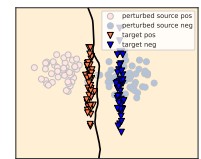

(a) Pre-trained model on source

(b) After DB towards target

Figure 3: Toy experiment of DB with ZO optimization.

### 3.2 EFFICIENT DOMAIN BRIDGING (EDB)

We realize that the ZO-based perturbation method applies *asynchronous* $X'_S$, which is always involved in local fine-tuning during a new round. Recall that the insight of DB is to explore how we perturb the source data so that the fine-tuned model can meet the adaptation expectations. We reorganize our learning objective for this adaptation task and propose Efficient Domain Bridging (EDB) as follows.

#### 3.2.1 OBJECTIVE FORMULATION

Denote the perturbation for the $i$-th source data point as $\delta_i$. We aim to learn sample-wise perturbation for source data such that by minimizing the training loss on perturbed source data we also achieve desired adaptation performance on unobserved support set. The whole objective can be written as

$$\theta^*(\delta) = \arg\min_\theta \underbrace{\frac{1}{n}\sum_{i=1}^n \ell(f(x_i + \delta_i; \theta), y_i)}_{\hat{\ell}(\delta;\theta)}, \tag{5}$$

$$\delta^* = \arg\min_{\forall x_i + \delta_i \in \mathcal{X}} V(\theta^*(\delta)), \tag{6}$$

where $\delta$ represents the set of sample-wise perturbation $\delta_i$, and $\hat{\ell}(\cdot)$ represents the impartial loss on perturbed source data. Note that $\delta$ differs from $\Delta$ in Eq. (3) which serves as a random direction and always occurs with a step factor $\epsilon$. The constraint in Eq. (6) can be implemented in a sample-wise style of Eq. (4). Hence, our new objectives can better describe the entire DB flow chart, as shown in Fig. 1, and the challenge now becomes how to solve $\theta^*(\cdot)$. Apparently, two objectives are dependent on each other, formulated as bi-level objectives (Sinha et al., 2017) but the outer one is nondifferentiable due to unobserved process in $V(\cdot)$. Generally, we can alternatively optimize $\theta$ and $\delta$ iteratively.

#### 3.2.2 APPROXIMATED SAMPLE-WISE PERTURBATION

Computing the closed-form solution to the instance-wise perturbation would be very difficult. We therefore propose first to approximate the gradient for each source data point $i$ and then project the updated source data onto the feasible data space $\mathcal{X}$ as indicated by Eq. (4). Drawing inspiration from the optimization techniques presented in Liu et al. (2018a), we derive the following proposition.

**Proposition 1.** *Let $\theta$ denote the current model parameters, $\xi$ is the learning rate for the training classifier with Eq. (5), and $\epsilon$ is a small scalar, the partial gradient of $V(\theta^*(\delta))$ with respect to $\delta_i$ in Eq. (6) can be approximated with a complexity of $O(d + |\theta|)$, that is*

$$\frac{\partial V(\theta^*(\delta))}{\partial \delta_i} \approx -\frac{\xi}{2\epsilon}\left(\frac{\partial\hat{\ell}(\delta;\theta^+)}{\partial\delta_i} - \frac{\partial\hat{\ell}(\delta;\theta^-)}{\partial\delta_i}\right) \tag{7}$$

*where $\theta^\pm = \theta \pm \epsilon\nabla_{\theta'}V(\theta')$ with $\nabla_{\theta'}V(\theta')$ computed by Eq. (1) and $\theta' = \theta - \xi\nabla_\theta\hat{\ell}(\theta)$.*

The proof of Proposition 1 can be referred to Appendix B. Note that Eq. (7) approximates the gradient w.r.t. perturbation $\delta_i$, and by repeatedly querying and updating we may reach the optimal $\delta^*$. During implementation, we can instead compute the gradient of $V(\theta)$ w.r.t. $x_i'$, making it have an aligned form with our expression in Section 3.1. We also performed this method on the toy examples and the adaptation results are left to Appendix D.2. For a detailed algorithmic procedure, one may refer to Algorithm 1.

We summarize that this proposed method fosters a stronger interaction between learning of model parameters $\theta$ and domain perturbations $\delta$. It applies ZO Optimization to a one-step unrolled model, enabling the gradient flow through the entire model back to the input under a bi-level optimization framework. With two levels of optimization – one focusing on the model parameters and the other on the perturbations – our approach provides a refined mechanism to optimize model performance when adapting to an unobserved target data.

---

**Algorithm 1** Efficient Domain Bridging (EDB)

**Require:** Initially provided model with parameter $\theta^0$, source data $(X_S, Y_S)$, target support set for evaluation $V(\cdot)$, number of iterations $T$, model parameter learning rate $\xi$, perturbation learning rate $\eta$, perturbation scale $\gamma$ and $\epsilon$

1: Initialize $\delta_i^0 = \mathbf{0}$ for all $x_i \in X_S$
2: **for** $t = 0, \ldots, T$ **do**
3:      Randomly partition $(X_S, Y_S)$ into mini-batches $\{(X_S^{(b)}, Y_S^{(b)})\}_{b=1}^B$ with corresponding $\{\delta^{(b)}\}_{b=1}^B$
4:      **for** each mini-batch $b = 1, \ldots, B$ **do**
5:          Sample $\mathbf{z} \sim \mathcal{N}(\mathbf{0}, \mathbf{I})$
6:          $V^{\pm} \leftarrow V(\theta^t \pm \gamma \mathbf{z})$
7:          $\hat{\mathbf{g}} \leftarrow \frac{V^+ - V^-}{2\epsilon} \mathbf{z}$
8:          $\theta^{\pm} \leftarrow \theta^t \pm \epsilon \hat{\mathbf{g}}$
9:          $\mathbf{g}^{\pm} \leftarrow \nabla_\delta \mathcal{L}(X_S^{(b)}, Y_S^{(b)}; \theta^{\pm})$
10:         $\mathbf{H} \leftarrow (\mathbf{g}^+ - \mathbf{g}^-)/(2\epsilon)$
11:         $\delta^{(b),t+1} \leftarrow \delta^{(b),t} - \eta \xi \mathbf{H}^{(b)}$
12:      **end for**
13:      $X_S \leftarrow \text{Proj}_{\mathcal{X}}\left(X_S + \delta^{t+1}\right)$
14:      $\theta^{t+1} \leftarrow \theta^t - \xi \nabla_\theta \mathcal{L}(X_S, Y_S; \theta^t)$
15: **end for**
16: **return** $\theta^{T+1}, \delta^{T+1}$

---

Based on the theoretical justification for stochastic implicit gradient estimation established in Zhang et al. (2021), we can derive a convergence guarantee for our EDB method under mild assumptions, which are naturally satisfied in our scenarios. We provide the proof in Appendix C.

## 4 EXPERIMENT

This section starts with an outline of the experimental setup, followed by a performance comparison with existing solutions. We also experimentally study the properties of the proposed domain bridging method across different dimensions.

### 4.1 EXPERIMENTAL SETUP

**Datasets**. Our methods are validated on five commonly adopted benchmarking datasets. **Office-31** (Saenko et al., 2010) and **Office-Home** (Venkateswara et al., 2017) are two relatively simple datasets with 3 and 4 domains, respectively. **PACS** (Li et al., 2017) and **VLCS** (Torralba & Efros, 2011) are two more complex datasets, both with 4 domains, which are also widely used in domain generalization tasks. **Amazon Review** (Blitzer et al., 2007) is binary text classification dataset with 4 domains. Following Jing et al. (2022), we split source data into 80% to train the source model and facilitate adaptation, and 20% for post-adaptation evaluation (e.g., one may wish to evaluate whether the adapted model maintains a good performance in source domain). Regarding the target data, we split them into two equal parts, a support set used for evaluation during adaptation and a holdout set to analyze the generalizability of the adapted model, following Li et al. (2023).

**Models**. The backbone models are selected following the convention of model adaptation literature. For Office-31 and Office-Home, we use ResNet-50 as in Liang et al. (2020). For the training process, we employ the SGD optimizer with a learning rate of 1e-2, a momentum of 0.9, a weight decay of 5e-4, and a batch size of 256. For PACS and VLCS, we use ResNet-18 as in Qu et al. (2023). We employ the SGD optimizer with a learning rate of 2e-3/1e-3 for PACS/VLCS respectively, a momentum of 0.9, and a batch size of 64. We conducted training over 30 epochs in the source domains. For the Amazon Review dataset, we leverage a BERT model for sequence classification (Li et al., 2022a), which involves an AdamW optimizer with a learning rate of 2e-5, utilizing batches of 32 samples for balanced training between positive and negative samples over 15 epochs.

Table 1: Classification accuracy (%) across different datasets.

| Method | Office-31 | Office-Home | PACS | VLCS | Amazon Review |
|--------|-----------|-------------|------|------|---------------|
| PRE | 73.00 | 52.95 | 53.63 | 62.77 | 89.88 |
| PGS | 74.56 | 54.75 | 57.36 | 65.36 | 91.13 |
| RSDV | 75.21 | 57.75 | 55.55 | 66.11 | 90.08 |
| EDB | **78.60** | **60.93** | **59.47** | **69.42** | **91.67** |
| | (0.58) | (0.74) | (0.41) | (0.45) | (0.40) |
| EDB* | 80.05 | 61.62 | 60.51 | 70.49 | 92.11 |
| | (0.36) | (0.90) | (0.45) | (0.42) | (0.47) |

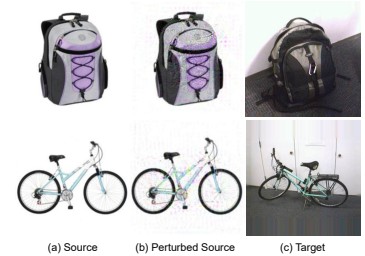

Figure 4: Examples of source, perturbed source, and target data on Office-31 dataset.

**Hyperparameters**. For $\xi$ and $\epsilon$, we follow the setting in (Liu et al., 2018a). Specifically, we set $\xi = 0.075$, $\gamma = 0.01$ and $\epsilon = 0.01/\|\nabla_{\theta'} V(\theta')\|_2$. We choose a sufficiently small learning rate $\eta = 0.01$ for the perturbation parameter $\delta$. For baseline comparisons, we use the source code of PGS and RSDV, and report their best performance from 10 independent runs for fair evaluation.

**Perturbation Application**. For image datasets, perturbations are directly applied to the pixel values. This approach involves adding noise with dimensions that match those of the images, which straightforwardly alters the pixel intensity values. In contrast, for text datasets, the input data is inherently discrete, which disables the direct application of continuous perturbations used for images. To address this, text inputs are first transformed into embeddings using BERT's text embedding layer. The perturbations are then added to these text embeddings. Consequently, for experimental purposes, we utilize these text embeddings as input for our algorithms rather than the raw text data. Additionally, to isolate the effects of our perturbations, the model adaptation process specifically excludes the embedding layer, ensuring that changes are focused on the upper layers of the model.

**Baselines**. We compare our method EDB against two important baselines, PGS (Li et al., 2023) and RSDV (Ghorbani & Zou, 2019), because they two are only solutions we can find from the literature, which can adapt a pre-trained model to unobserved target data based on model performance. In addition to including the performance of the pre-trained model (dubbed as "PRE" for short) as a reference, we apply the exact gradient when computing $\nabla_\theta V(\theta)$ in EDB, thus represented by EDB* subsequently. This method helps to understand how the precision of the estimated gradient affects the adaptation, serving the boundary performance of our method. The best results of the methods compared that exclude EDB* are marked in bold.

## 4.2 ADAPTATION PERFORMANCE COMPARISON

**Image Classification.** The adaptation performance across various image classification datasets is presented in Tab.1. Our proposed method, EDB, consistently outperforms baseline approaches for all datasets. We present the detailed experimental results in Appendix D.3, which demonstrate that our method achieves superior performance over baselines on almost all domain transfer pairs. This consistent superiority demonstrates the robustness of our approach across diverse visual domains. Specifically, on the Office-31 and Office-Home dataset, EDB achieves an improvement of 3% in average accuracy. On the more challenging PACS and VLCS dataset, which contains more data samples and greater visual diversity, our method still demonstrates an impressive improvement over baseline methods, highlighting the effectiveness of our domain bridging technique across complex visual domains. Furthermore, EDB* outperforms all other methods by a significant margin, indicating the potential for further advancement in the current approach with a more precise estimator.

**Text Classification.** The results for text classification on the Amazon review dataset are also presented in Tab.1, where EDB still outperforms baseline methods. Although the improvement seems marginal, it is comparable to that reported in Li et al. (2022a). Moreover, our improvements are consistent across all 12 domain transfer pairs and statistically significant. Importantly, this suggests that adding perturbations to text embeddings is effective. Therefore, our EDB method is not restricted to operating on raw data alone but is equally applicable to intermediate representations, such as text embeddings as shown here, and other encoder outputs in various models. By applying perturbations at different stages within the model's pipeline, EDB can flexibly adapt to shifts across a wide range of data modalities and architectures.

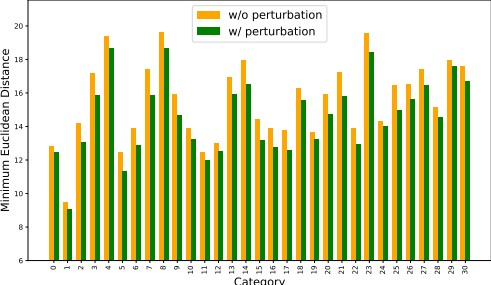

Figure 5: Minimum euclidean distance between source and target data with and without perturbation over embedding space on Office-31 dataset.

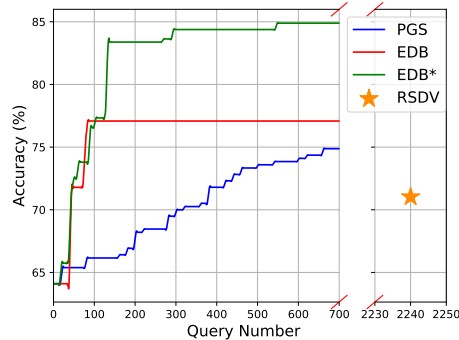

Figure 6: Query efficiency comparison on Office-31 (A-W).

One important thing we claim about PGS in Sec. 2.1 is that it overfits to support set and generalizes poorly on holdout set. We show that EDB effectively reduces this overfitting behavior from the smaller support-holdout accuracy gaps on Office-31 dataset compared to PGS, as shown in Tab. 2.

**Multi-source domain adaptation**. Beyond single-domain adaptation, we also conducted multi-source domain adaptation experiments. In this setting, we combine any three domains as multi-source and adapt to the remaining domain as target. This configuration closely resembles multi-

Table 2: Absolute difference between support and holdout accuracy on Office-31. Lower values indicate less overfitting.

|  | A-W | A-D | W-A | W-D | D-A | D-W | Average |
|---|---|---|---|---|---|---|---|
| PGS | 3.89 | 11.90 | 2.41 | 0.37 | 2.66 | 2.78 | 4.00 |
| EDB | 0.69 | 0.66 | 1.17 | 0.20 | 0.97 | 0.53 | 0.70 |

source domain generalization (DG), with the key distinction being the availability of target domain feedback during the adaptation process. We implement our EDB method in this multi-source setting using the PACS dataset and benchmark against established DG approaches. Our results in Appendix E demonstrate that EDB achieves at least 4.5% improvement in average accuracy compared to existing DG methods, highlighting the substantial benefit of incorporating target domain feedback to guide the adaptation process. We also provide comparisons to single-domain DG in Appendix E.

### 4.3 UNDERSTANDING THE "BRIDGE"

We take a closer look at the perturbation to further understand the concept of DB. For this purpose, we use Office-31 as an example dataset, with a focus on the (A-W) adaptation unless otherwise specified.

**Perturbation Visualization.** Have we transformed the source data points into target-like points through input perturbations? Fig. 4 presents selected images from the *bag* and *bike* classes in the Office-31 dataset, from which we can see that the perturbations do not make the perturbed source data visually resemble the target data. This observation alleviates concerns that our method might expose target information by constructing a bridge between two domains. In addition, recent research (Beetham et al., 2023) has shown that it is unnecessary to replicate exact target features. Instead, the focus should be on learning target-related discriminative information that enhances the classification performance of the model assessment.

**Representation View.** In addition to the input space, we quantify the distance between the source and target data in the embedding space. To achieve this, we utilize the Minimum Euclidean Distance (MED), which identifies the nearest target sample from the same class and computes the L2 distance, and thus is more sensitive to perturbations. Results are shown in Fig. 5. We observe that across all classes, the distances are clearly reduced after applying the perturbations, demonstrating that the perturbations obtained by our method effectively draw the features of source data closer to the unobserved target data. Note that our findings are not limited to MED; please refer to Appendix H.

**Performance Preserving.** Although preserving model performance on source data is not a required property for evaluation-based model adaptation tasks, we are curious whether fine-tuning on the perturbed source data affects the model's generalization on the source domain. Tab.3 summarizes the classification performance

Table 3: Post-adaptation classification accuracy (%) on source domain on Office-31 dataset.

|  | A-W | A-D | W-A | W-D | D-A | D-W |
|---|---|---|---|---|---|---|
| Before | 99.87 | | 100.00 | | 100.00 | |
| After | 97.37 | 97.44 | 98.74 | 99.24 | 99.20 | 98.80 |

changes before and after fine-tuning. The results show that the performance drop is limited to within 2%, even without adding an additional term to explicitly fit the benign source data.

### 4.4 ROBUSTNESS ANALYSIS

**Limited target data**. In practical scenarios, data holders may only be willing to share a small subset of their data for evaluation purposes due to privacy leakage concerns. To address such limitations, our method does not require extensive support data. Specifically, EDB remains effective with as little as 20% of the support data, with detailed experimental results presented in Appendix F, although more data will bring improved adaptation performance.

**Noisy data**. In practical scenarios, the feedback from target domains may be imperfect due to various factors such as data corruption or label noises. These imperfections can lead to less accurate performance feedback, thus making adaptation a more challenging task. To evaluate the robustness of our method under such conditions, we conducted experiments with simulated noisy feedback. Specifically, we added varying levels of Gaussian noise to the feedback values. Our results demonstrate that while EDB's performance gradually degrades with increasing noise levels, it remains within an acceptable range and consistently outperforms baseline methods even when they use noise-free feedback. We provide the results in Appendix D.4.

### 4.5 EFFICIENCY INVESTIGATION

Recall that query efficiency is critical for evaluation-based model adaptation, as it impacts communication cost, evaluation cost, and privacy cost. Specifically, a higher number of queries implies more instances where model providers must send their model (or model parameters) to clients, perform forward inferences, and face increased risks of data leakage. Fig. 6 illustrates the classification performance of the compared methods on the support set (Office-31 A-W). We report the best historical performance for each query, resulting in a monotonically increasing accuracy curve. From this figure, we make following observations: (1) Both EDB and EDB$^*$ converge faster and achieve a higher accuracy than PGS. (2) The gap between EDB and EDB$^*$ during tuning remains evident, suggesting that the accuracy could be further improved with a more precise estimator in our method. (3) RSDV requires the highest number of queries, equivalent to the size of the source training data, which is significantly larger than that of other methods, while achieving the worst performance on support set.

While query efficiency addresses the communication burden with the target domain, scalability to large source datasets is equally important for practical deployment. Our EDB method learns sample-wise perturbations that can be computed in parallel within each batch, similar to the parallelization strategies used in adversarial robustness training (Madry, 2017) where large datasets are routinely handled. The gradient computation for perturbations (as shown in Equation 7) can be efficiently parallelized across source samples, enabling our method to scale gracefully to large-scale source datasets. This computational efficiency, combined with EDB's rapid convergence, results in significantly reduced training time compared to baseline methods. For detailed computational efficiency and convergence analysis, please refer to Appendix G.

## 5 CONCLUSION

In this work, we revisit the evaluation-based model adaptation problem and analyze the limitations of existing solutions. Recognizing the benefits of incorporating source data during adaptation, we introduced the concept of domain bridging. Unlike zeroth-order optimization, which may be unreliable for learning sample-wise input perturbations, we proposed a new efficient method to learn the domain feature differences. Experimental results on image and text classification tasks demonstrate that our method effectively reduces the gap between source and unobserved target domains. Future directions for this work include exploring more accurate zeroth-order gradient estimation techniques to further reduce the approximation error in our framework. Furthermore, investigating adaptive perturbation mechanisms that dynamically adjust to different data distributions could further enhance the versatility and performance of our method. Training a parameterized perturbation generator is also worth exploring, especially when the query budget is sufficiently large. These advancements would push the boundaries of privacy-preserving model adaptation and solidify domain bridging as a practical and scalable solution for real-world applications.

## ETHICS STATEMENT

Our research on domain bridging for evaluation-based model adaptation raises no ethical concerns. The method does not involve human subjects or experiments. We do not collect, store, or process any personal or sensitive data. The approach is designed to enhance privacy by enabling model adaptation without direct access to target data, thereby protecting proprietary and sensitive information. All experiments were conducted on publicly available benchmark datasets following standard evaluation protocols. Our code will be made available to support reproducibility and advance research in privacy-preserving machine learning.

## REPRODUCIBILITY STATEMENT

The code is provided in the supplementary materials. The paper provides sufficient instructions for reproducing the experimental results of the paper in Sec4.1. All datasets used in the research are publicly available.

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

We emphasize our research position in Appendix A, particularly contrasting our evaluation-based DA approach with traditional DA methods and approaches from adjacent fields. Appendices B and C provide proofs for Proposition 1 and convergence guarantees, respectively. Appendix D contains comprehensive experimental details, especially detailed evaluation of EDB across five datasets. Appendix E compares our method with domain generalization (DG) approaches. Appendix F studies the impact of support set size on EDB performance. Appendix G provides empirical computational time analysis, and Appendix H demonstrates feature similarity changes using cosine distance.

## A  RESEARCH POSITION

We situate our work within the field of Domain Adaptation (DA). However, our method introduces a new paradigm in the field – enabling adaptation by performance feedback from the target domain, without accessing target features or labels. This is particularly important in scenarios where model developers cannot directly access sensitive target data, such as Machine Learning as a Service (MLaaS) (Philipp et al., 2020) in healthcare. Our core technique is using feedback to estimate potential domain differences via an approximate bi-level optimization, guiding the fine-tuned model to progressively adapt to the unseen target data.

Recall that in the evaluation-based model adaptation problem, the candidate model will be evaluated in the support set multiple times. Beyond inference computation cost, we highlight that privacy leakage in this case is more likely in membership than data content (Zelikman et al., 2023), because it is difficult to recover any data information from average performance alone. Generally, membership privacy can be addressed using techniques such as differential privacy (Dwork et al., 2014), and (Li et al., 2023) also experimentally demonstrated that the precision of feedback (the noise for privacy will be on the feedback) only slightly impacts the adaptation performance. Clients can alternatively select representative target data points to share to fine-tune (Zhang et al., 2022). It is uncertain which choice would be better because different scenarios may prioritize different types of data information.

Since there is a great deal of literature in DA, we clarify the difference between our paradigm and others by summarizing the closely related problem setups, as shown in Tab.4. Unsupervised DA (Ganin & Lempitsky, 2015) is the most widely studied, where a source model may or may not be provided, and target data are typically unlabeled. Source-free DA (Li et al., 2024) solves a similar problem but with no source data provided during adaptation. We notice some source-free DA research (Wang et al., 2020) claimed the privacy issue of source data, and in this view, Evaluation-based DA can be regarded as a dual research problem. Few-shot DA shares the same motivation as us, but they tend to use some of the target data for fine-tuning. Unlike (Li et al., 2023), we emphasize the necessity of source data in Evaluation-based DA because it provides valuable information for model development in the absence of target data.

Meanwhile, there are also related approaches from adjacent fields that involve data modifications. For example, learning perturbations on source data based on feedback connects our work to **Adversarial Learning** research (Biggio et al., 2013; Madry, 2017), where adversarial perturbations are constrained by a small $L_p$-norm value. Please note that adversarial perturbation is derived by a sample-wise query and thus the learned perturbation is more accurate, even for black-box model scenarios, while in our work we utilize a series of set-based performance to update the sample-wise perturbation. In addition, from Equation 7 we can see that the gradient of domain perturbation is more complex than that of adversarial perturbation, which often involves first-order information. **Data Augmentation** techniques (Shorten & Khoshgoftaar, 2019) typically apply trainable or non-trainable data transformations without using model performance feedback on the target domain. Our method uniquely combines data perturbation with target domain performance feedback to guide adaptation, representing a novel intersection of these approaches.

Table 4: Comparison of data and model requirements for different problem setups. A $-$ indicates "not required".

|  | Source Data | Source Model | Target Data |
| --- | --- | --- | --- |
| Standard Fine-Tuning | $-$ | ✓ | $X_T, Y_T$ |
| Unsupervised DA | ✓ | $-$ | $X_T$ |
| Source-free DA | ✗ | ✓ | $X_T$ |
| Few-shot DA | ✗ | ✓ | $\{x_T, y_T\}^k$ |
| Evaluation-based DA | ✓ | ✓ | $V(X_T, Y_T)$ |

## B   PROOF OF PROPOSITION 1

*Proof.* Let $\theta' = \theta - \xi\nabla_\theta\hat{\ell}(\theta)$, which is a one-step update to the current model parameters. We have

$$\frac{\partial V(\theta^*(p))}{\partial \delta_i} \approx \frac{\partial V(\theta - \xi\nabla_\theta\hat{\ell}(\theta))}{\partial \delta_i} \tag{8}$$

$$= \frac{\partial V(\theta')}{\partial \delta_i} \tag{9}$$

$$= \frac{\partial V(\theta')}{\partial \theta'}\frac{\partial \theta'}{\partial \delta_i} \tag{10}$$

$$= \frac{\partial V(\theta')}{\partial \theta'}\left(-\xi\frac{\partial^2\hat{\ell}(\theta)}{\partial \delta_i\partial\theta}\right) \tag{11}$$

$$\approx -\frac{\xi}{2\epsilon}\left(\frac{\partial\hat{\ell}(\delta;\theta^+)}{\partial \delta_i} - \frac{\partial\hat{\ell}(\delta;\theta^-)}{\partial \delta_i}\right) \tag{12}$$

where $\theta^\pm = \theta \pm \epsilon\nabla_{\theta'}V(\theta')$.

$\square$

Here we provide key steps only, and one can refer to (Liu et al., 2018a) for more detailed steps about this gradient estimation.

## C   PROOF OF CONVERGENCE

**Theorem 1** (Convergence of EDB). *Under mild assumptions that: (i) $V$ is $L$-smooth and bounded below, (ii) the training loss $\ell(\delta;\theta)$ is $\mu$-strongly convex in $\theta$ and has bounded Hessian, (iii) $\eta$ is sufficiently small to satisfy standard diminishing conditions, the proposed Efficient Domain Bridging (EDB) method converges in expectation to a stationary point.*

*Proof.* Following the bi-level optimization framework and adapting techniques from Zhang et al. (2021), we establish convergence as follows.

Let $\theta^*(\delta)$ denote the solution to the inner optimization problem, and $\tilde{V}$ be our zeroth-order approximation. Let $\delta_t$ denote the perturbation at iteration $t$.

From Proposition 1, the gradient approximation error satisfies:

$$\|\nabla_\delta V(\theta^*(\delta)) - \nabla_\delta\tilde{V}(\theta^*(\delta))\| \leq \frac{C}{\epsilon}(1-\xi\mu)^K \tag{13}$$

where $C$ is a constant depending on Lipschitz constants, and the error decreases exponentially with $K$.

Since $V$ is $L$-smooth, we have that $\nabla_\delta V$ is $L$-Lipschitz continuous. Following Zhang et al. (2021), the iterative update yields:

$$\mathbb{E}[V(\theta^*(\delta_{t+1}))] \leq \mathbb{E}[V(\theta^*(\delta_t))] - \eta\left(1 - \frac{C(1-\xi\mu)^K}{\epsilon\|\nabla_\delta\tilde{V}\|}\right)\mathbb{E}[\|\nabla_\delta\tilde{V}\|^2] + \frac{L\eta^2}{2}\mathbb{E}[\|\nabla_\delta\tilde{V}\|^2] \tag{14}$$

Rearranging terms:

$$\mathbb{E}[V(\theta^*(\delta_{t+1}))] \leq \mathbb{E}[V(\theta^*(\delta_t))] - \eta\left(1 - \frac{L\eta}{2} - \frac{C(1-\xi\mu)^K}{\epsilon\|\nabla_\delta\tilde{V}\|}\right)\mathbb{E}[\|\nabla_\delta\tilde{V}\|^2] \tag{15}$$

Choosing $\eta$ sufficiently small such that $\eta < \frac{2}{L}$ and the term $\left(1 - \frac{L\eta}{2} - \frac{C(1-\xi\mu)^K}{\epsilon\|\nabla_\delta\tilde{V}\|}\right) > 0$, and telescoping over $T$ iterations:

$$\sum_{t=0}^{T-1} \eta \mathbb{E}[\|\nabla_\delta \tilde{V}(\theta^*(\delta_t))\|^2] \leq V(\theta^*(\delta_0)) - \mathbb{E}[V(\theta^*(\delta_T))] \tag{16}$$

Since $V$ is bounded below and using diminishing step sizes satisfying $\sum_t \eta_t = \infty$ and $\sum_t \eta_t^2 < \infty$, we obtain:

$$\liminf_{t\to\infty} \mathbb{E}[\|\nabla_\delta \tilde{V}(\theta^*(\delta_t))\|] = 0 \tag{17}$$

Therefore, the EDB algorithm converges in expectation to a stationary point. □

## D    EXPERIMENTS RESULTS

### D.1    EXPERIMENT OF DB WITH ZO OPTIMIZATION

We provide the experimental results of DB with ZO optimization, as described in Section 3.1, on Office-31 (A-W). Fig. 7 suggests that this method struggles to facilitate effective adaptation to the target data. The failure is likely attributable to the unreliable gradient estimation produced by ZO, particularly when dealing with high-dimensional data, which leads to inaccurate perturbation directions.

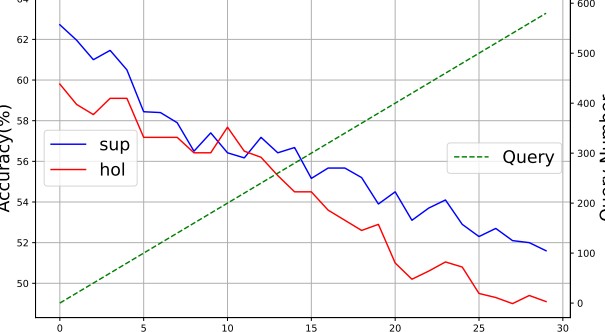

Figure 7: Accuracy curves of DB with ZO optimization on support set and holdout set, with the corresponding number of queries.

### D.2    TOY EXPERIMENT OF EDB

We present the outcomes of our EDB method on the toy experiment as discussed in Section 3.1. Fig. 8b demonstrates that the pre-trained model adapts well to the target data. The sample-wise perturbation further enhances the effectiveness of the adaptation.

### D.3    DETAILED RESULTS OF MAIN EXPERIMENTS

In this section, we provide detailed experimental data for our experiments. From Tables 5, 6, 7, 8 and 9, we can observe that our EDB method achieves consistent improvements across the vast majority of domain transfer pairs.

### D.4    ROBUSTNESS USING NOISY DATA

We evaluate our method's robustness to noisy feedback using the Office-31 A-W task. To simulate realistic scenarios where target data may be imperfect, we corrupt the validation feedback with varying levels of Gaussian noise $\mathcal{N}(0, \sigma)$. As shown in Tab. 10, our EDB approach still outperforms baseline methods even when they use noise-free feedback.

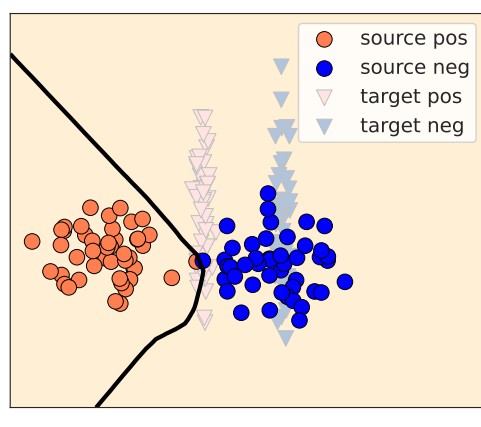 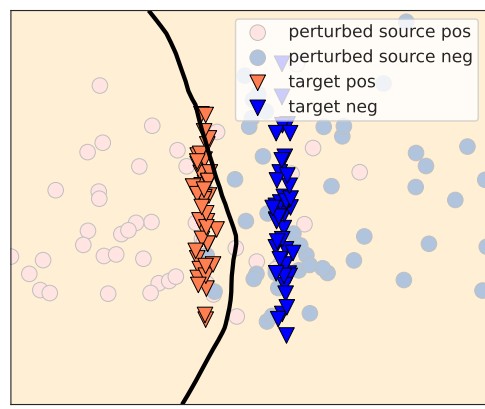



(a) Pre-trained model on source      (b) After EDB towards target

Figure 8: Toy experiment of EDB



Table 5: Classification accuracy (%) on Office-31 dataset.

| Method | A-W | A-D | W-A | W-D | D-A | D-W | Average |
|--------|-----|-----|-----|-----|-----|-----|---------|
| PRE | 64.10 | 67.10 | 57.91 | 98.45 | 57.33 | 93.11 | 73.00 |
| PGS | 67.65 | 67.65 | 62.02 | 99.22 | 58.49 | 92.35 | 74.56 |
| RSDV | 70.86 | 75.29 | 59.05 | 98.82 | 55.87 | 91.36 | 75.21 |
| EDB | **73.37** | **75.29** | **63.51** | **99.44** | **63.93** | **96.05** | **78.60** |
|  | (1.34) | (0.77) | (0.48) | (0.22) | (1.54) | (0.69) | (0.58) |
| EDB* | 76.30 | 79.22 | 63.73 | 99.60 | 64.21 | 97.24 | 80.05 |
|  | (1.15) | (0.87) | (0.34) | (0.40) | (0.59) | (0.65) | (0.36) |

Table 6: Classification accuracy (%) on Office-Home dataset.

| Method | A-C | A-P | A-R | C-A | C-P | C-R | P-A | P-C | P-R | R-A | R-C | R-P | Average |
|--------|-----|-----|-----|-----|-----|-----|-----|-----|-----|-----|-----|-----|---------|
| PRE | 42.47 | 56.68 | 68.01 | 43.63 | 52.67 | 54.06 | 40.75 | 37.16 | 65.37 | 57.73 | 46.44 | 70.47 | 52.95 |
| PGS | 45.91 | 61.35 | 69.67 | 44.45 | 55.90 | 55.97 | 42.09 | 38.77 | 64.25 | 55.06 | 49.91 | 73.64 | 54.75 |
| RSDV | 43.00 | 59.34 | 69.95 | 50.49 | 57.77 | 62.39 | 49.27 | 41.77 | **71.63** | 63.87 | 47.55 | 75.92 | 57.75 |
| EDB | **48.41** | **65.24** | **74.71** | **52.61** | **60.96** | **65.12** | **49.65** | **49.35** | 71.58 | **65.66** | **50.91** | **77.39** | **60.93** |
|  | (0.89) | (0.75) | (0.45) | (0.86) | (1.21) | (0.83) | (0.67) | (0.81) | (0.78) | (0.19) | (0.81) | (0.57) | (0.74) |
| EDB* | 48.95 | 66.03 | 74.90 | 54.40 | 61.30 | 65.66 | 49.76 | 50.33 | 72.36 | 66.39 | 51.67 | 77.66 | 61.62 |
|  | (1.06) | (1.26) | (0.83) | (0.96) | (0.42) | (0.99) | (0.91) | (0.86) | (0.65) | (0.66) | (1.35) | (0.86) | (0.90) |

Table 7: Classification accuracy (%) on PACS dataset.

| Method | P-A | P-C | P-S | A-P | A-C | A-S | C-P | C-A | C-S | S-P | S-A | S-C | Average |
|--------|-----|-----|-----|-----|-----|-----|-----|-----|-----|-----|-----|-----|---------|
| PRE | 69.51 | 28.60 | 32.06 | 96.00 | 56.64 | 41.00 | 86.31 | 66.58 | 59.01 | 34.93 | 28.40 | 44.50 | 53.63 |
| PGS | 72.26 | 31.02 | 32.60 | 97.13 | 61.69 | 52.67 | 88.26 | 69.82 | 64.66 | 38.53 | 31.26 | 48.37 | 57.36 |
| RSDV | 72.56 | 29.82 | 31.79 | 96.41 | 60.15 | 48.19 | 88.50 | 67.29 | 59.13 | 36.05 | 28.42 | 48.29 | 55.55 |
| EDB | **74.02** | **31.23** | **35.18** | **97.84** | **64.25** | **56.74** | **89.70** | **71.58** | **68.04** | **39.40** | **34.67** | **51.02** | **59.47** |
|  | (0.58) | (0.34) | (0.44) | (0.25) | (0.56) | (0.33) | (0.52) | (0.43) | (0.24) | (0.43) | (0.52) | (0.23) | (0.41) |
| EDB* | 74.51 | 31.78 | 36.29 | 98.02 | 65.36 | 57.69 | 90.30 | 72.80 | 70.02 | 40.51 | 36.12 | 52.73 | 60.51 |
|  | (0.61) | (0.35) | (0.36) | (0.21) | (0.44) | (0.46) | (0.71) | (0.64) | (0.58) | (0.25) | (0.55) | (0.20) | (0.45) |

Table 8: Classification accuracy (%) on VLCS dataset.

| Method | C-L | C-S | C-V | L-C | L-S | L-V | S-C | S-L | S-V | V-C | V-L | V-S | Average |
|---|---|---|---|---|---|---|---|---|---|---|---|---|---|
| PRE | 54.10 | 48.90 | 53.30 | 88.00 | 56.50 | 61.80 | 34.50 | 60.90 | 60.40 | 98.60 | 59.00 | 77.20 | 62.77 |
| PGS | 55.31 | 51.74 | 55.69 | 88.00 | 63.45 | 62.40 | 38.78 | 64.42 | 63.41 | 98.95 | 62.66 | 79.56 | 65.36 |
| RSDV | 57.34 | 52.96 | 56.63 | 88.56 | 62.45 | 63.41 | 42.27 | 66.78 | 63.00 | 99.10 | 61.63 | 79.16 | 66.11 |
| EDB | **65.79** | **55.62** | **59.47** | **88.73** | **66.26** | **65.09** | **45.55** | **71.05** | **70.57** | **99.30** | **64.78** | **80.79** | **69.42** |
| | (0.47) | (0.48) | (0.59) | (0.12) | (0.56) | (0.68) | (0.33) | (0.64) | (0.70) | (0.17) | (0.44) | (0.23) | (0.45) |
| EDB* | 66.72 | 57.73 | 59.88 | 88.85 | 67.30 | 68.24 | 46.90 | 71.39 | 71.14 | 99.30 | 66.90 | 81.55 | 70.49 |
| | (0.58) | (0.41) | (0.65) | (0.20) | (0.34) | (0.75) | (0.47) | (0.55) | (0.56) | (0.10) | (0.36) | (0.16) | (0.42) |

Table 9: Classification accuracies (%) on Amazon Review holdout dataset.

| Method | B-D | B-E | B-K | D-B | D-E | D-K | E-B | E-D | E-K | K-B | K-D | K-E | Average |
|---|---|---|---|---|---|---|---|---|---|---|---|---|---|
| PRE | 91.15 | 90.65 | 92.00 | 91.2 | 89.10 | 91.00 | 84.80 | 87.35 | 92.60 | 89.05 | 88.10 | 91.50 | 89.88 |
| PGS | 91.80 | **92.40** | 92.50 | **92.10** | 90.50 | 92.70 | 87.80 | 88.70 | 93.50 | 89.40 | 89.50 | 92.70 | 91.13 |
| RSDV | 88.30 | 91.00 | 92.40 | 90.20 | 88.70 | 92.40 | 88.80 | 86.00 | **94.50** | 88.80 | 87.00 | 92.90 | 90.08 |
| EDB | **92.48** | 92.30 | **93.20** | 91.56 | **91.12** | **93.00** | **89.14** | **89.30** | 94.14 | **89.78** | **90.38** | **93.66** | **91.67** |
| | (0.36) | (0.37) | (0.47) | (0.40) | (0.22) | (0.37) | (0.46) | (0.60) | (0.30) | (0.50) | (0.42) | (0.38) | (0.40) |
| EDB* | 93.24 | 93.04 | 93.50 | 92.38 | 91.54 | 93.22 | 90.04 | 88.88 | 94.40 | 90.38 | 90.50 | 94.18 | 92.11 |
| | (0.50) | (0.65) | (0.37) | (0.48) | (0.34) | (0.56) | (0.63) | (0.45) | (0.27) | (0.34) | (0.66) | (0.41) | (0.47) |

Table 10: Robustness evaluation with noisy feedback on Office-31 A-W task. We add Gaussian noise $\mathcal{N}(0, \sigma)$ to the feedback, and $\sigma = 0$ suggests no noise.

| $\sigma$ | 0 | 0.01 | 0.03 | 0.05 |
|---|---|---|---|---|
| PRE | 64.10 | – | – | – |
| PGS | 67.65 | – | – | – |
| RSDV | 70.86 | – | – | – |
| EDB | 73.37 | 73.01 | 72.48 | 71.52 |

# E  COMPARISONS TO DOMAIN GENERALIZATION METHODS

## E.1  MULTI-SOURCE DOMAIN GENERALIZATION

First, we compare EDB with multi-source domain generalization (DG) methods in Tab.11. Here, each column represents a target domain, where the other three domains are used to train the pre-trained model. We use the DG results provided by Wang et al. (2022). For fair comparison, we strictly follow their experimental setups on the PACS dataset. We can observe the significant improvements brought by our method through utilizing target data feedback.

## E.2  SINGLE-SOURCE DOMAIN GENERALIZATION

Now we compare our EDB method to single source domain generalization (DG) methods. Specifically, we follow the experimental setups and directly use the results provided by Qu et al. (2023). Note that each column represents the average performance when using one source domain to generalize to the other three target domains. For example, the V column indicates that we train the source model on domain V and then evaluate the average accuracy across the other three domains. From Tab.12, we can observe that EDB consistently outperforms DG methods by a significant margin. Again, this demonstrates the substantial improvement achieved by leveraging indirect feedback from target data, even without direct access to target samples during training.

Table 11: Multi-source EDB on PACS dataset

| Method | A | C | P | S | Average |
|---|---|---|---|---|---|
| ERM | 77.0 | 74.5 | 95.5 | 77.8 | 81.2 |
| DANN (Ganin et al., 2016) | 78.7 | 75.3 | 94.0 | 77.8 | 81.4 |
| CORAL (Sun & Saenko, 2016) | 77.7 | 77.0 | 92.6 | 80.5 | 82.0 |
| Mixup (Zhang et al., 2017) | 79.1 | 73.4 | 94.4 | 76.7 | 80.9 |
| RSC (Huang et al., 2020) | 79.7 | 76.1 | 95.6 | 76.6 | 82.0 |
| GroupDRO (Sagawa et al., 2019) | 76.0 | 76.0 | 91.2 | 79.0 | 80.6 |
| ANDMask (Parascandolo et al., 2020) | 76.2 | 73.8 | 91.6 | 78.0 | 79.9 |
| **EDB** | **86.1** | **78.9** | **97.4** | **83.5** | **86.5** |

Table 12: Single-source EDB on VLCS dataset

| Method | V | L | C | S | Average |
|---|---|---|---|---|---|
| ERM | 76.72 | 58.86 | 44.95 | 57.71 | 59.56 |
| ERM w/ MAD | 76.21 | 67.97 | 46.55 | 61.04 | 62.95 |
| ACVC (Cugu et al., 2022) | 76.15 | 61.23 | 47.43 | 60.18 | 61.25 |
| ACVC w/ MAD | 76.15 | 69.36 | 48.04 | 61.74 | 63.82 |
| DSU (Li et al., 2022b) | 76.93 | 69.20 | 46.54 | 58.36 | 62.76 |
| DSU w/ MAD | 76.99 | 70.85 | 44.78 | 62.23 | 63.71 |
| **EDB** | **81.62** | **73.36** | **60.29** | **62.39** | **69.42** |

## F  SUPPORT SET SIZE INVESTIGATION

In practical scenarios, access to target data may be further limited due to privacy concerns, high acquisition costs, or proprietary constraints. Therefore, we investigate how our EDB method performs under different support set sizes to evaluate its effectiveness in limited data scenarios.

Fig. 9 illustrates the relationship between support set size, model performance, and computational efficiency. We conducted experiments on the Office-31 dataset (A-W) by varying the support set size from 20% to 100% of the available support set while maintaining the same evaluation protocol on the holdout set. The results demonstrate that our method exhibits robust performance even with substantially reduced support data. When reducing the support set to just 20% (fewer than 100 samples), we observe only a 5.37% performance degradation compared to using the full support set. This minimal degradation confirms that Domain Bridging can effectively capture the essential domain shift patterns with limited data.

Beyond maintaining performance with limited data, reducing the support set size brings significant computational benefits. As shown in Fig. 9, the time reduction scales nearly linearly with the decrease in support set size. At 20% support set size, we observe approximately 40% reduction in training time compared to using the full support set.

This efficiency gain is particularly valuable in scenarios when the dataset is large or the computational resources are constrained. The balance between performance and computational efficiency is optimized at around 40-60% support set size, where we maintain approximately 97% of the full performance while reducing training time by 20-30%. These findings have important implications for real-world applications. When target domain data owners are reluctant to use their complete datasets due to privacy concerns, our method can still operate effectively with a small subset of their data. Additionally, in scenarios where rapid adaptation is required, users can strategically reduce the support set size to achieve faster adaptation with minimal performance sacrifice.

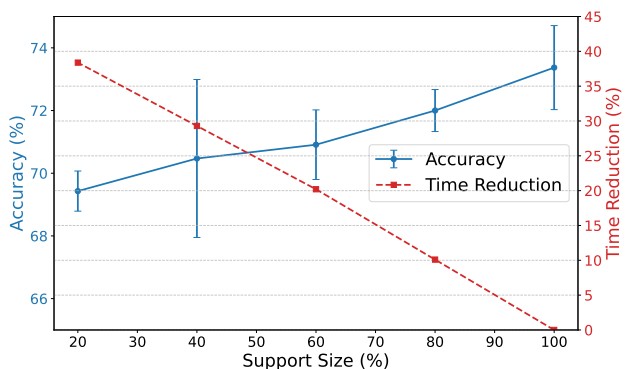

Figure 9: Support Set Size Impact on Model Performance and Training Efficiency

# G   COMPUTATIONAL EFFICIENCY AND CONVERGENCE ANALYSIS

As implied by Proposition 1, the time complexity per step (including both upper-level and lower-level updates) of our method, which depends upon the size of the source data, is relatively higher than baseline methods. While concerns may be raised that our method requires more computational time, it in fact significantly reduces the computational time due to faster convergence compared to baseline methods, as shown in Fig. 6.

Tab.13 presents a computational time comparison across tasks with different source data sizes. To ensure fair comparison and remove the effect of support set size variation, we use partial support sets of roughly the same sizes across all tasks. The results demonstrate that EDB achieves remarkable efficiency, requiring only about 40% of PGS's computational time and less than 10% of RSDV's time across all tasks. This demonstrates that EDB scales well with various dataset sizes, making it suitable for large-scale dataset scenarios while maintaining both accuracy and efficiency advantages.

Table 13: Computational time comparison (in seconds) across different tasks.

|  | Office-31 W-A | Office-Home A-C | Office-Home C-A |
| --- | --- | --- | --- |
| Source Data Size | 636 | 1942 | 3492 |
| Target Data Size | 1127 | 1091 | 1092 |
| PGS | 273 | 714 | 981 |
| RSDV | 835 | 2339 | 3420 |
| **EDB** | **88** | **245** | **412** |

Fig. 10 presents the convergence curves across corresponding tasks. According to our experimental results, although EDB requires more time as the source data size increases, it typically achieves successful convergence after three to four epochs of full perturbation over the source data. This rapid convergence leads EDB to be significantly faster than other baseline approaches, as demonstrated in Tab.13.

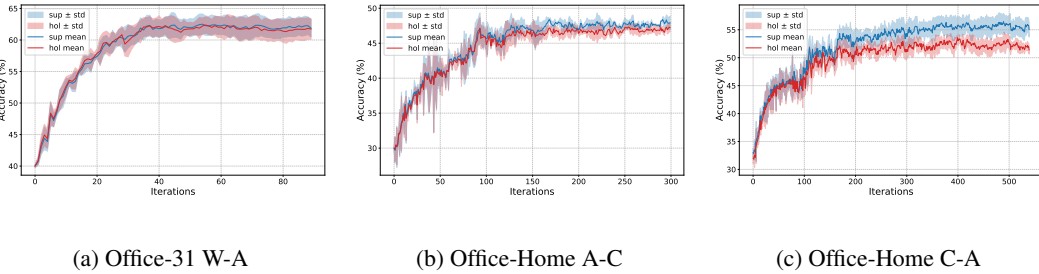

(a) Office-31 W-A            (b) Office-Home A-C            (c) Office-Home C-A

Figure 10: Convergence curves across different tasks

# H  REPRESENTATION VIEW WITH COSINE DISTANCE

We further examine the feature similarity using cosine distance in the embedding space. As shown in Fig. 11, the minimum cosine distance to the nearest target sample from the same class decreases after perturbation across all classes. This confirms that our method effectively reduces the distance between the source and target representation, consistent across different distance measures.

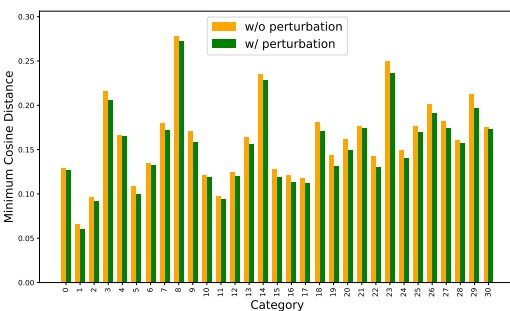

Figure 11: Minimum Cosine distance between source and target data with and without perturbation over embedding space on Office-31 dataset.

# I  LLM USAGE DECLARATION

We used large language models solely for grammatical corrections and language polishing of our manuscript. All research ideas, theoretical contributions, experimental design, implementation, data analysis, and interpretation of results were conducted entirely by the authors. The LLMs were used exclusively as a writing aid to improve clarity and correct grammatical errors in the final presentation of our work. No scientific content, analysis, or conclusions were generated or influenced by LLMs.

