# OpenReview forum: "Domain Bridging: Enabling Adaptation without Peeking at Target Data"
_ICLR.cc/2026/Conference — ICLR 2026 Conference Withdrawn Submission_

### Official Review · Reviewer_Eamd · 2025-10-30

**Soundness:** 3
**Presentation:** 3
**Contribution:** 3
**Rating:** 4
**Confidence:** 3

**Summary:**

The paper tackles evaluation-based domain adaptation when target data are inaccessible. Instead of updating model weights directly from performance-only feedback or using a 2-stage 'value then retrain' method, the authors propose Domain Bridging, learn sample-wise perturbations of the source data that make standard fine-tuning on the perturbed source behave as if training on the unseen target. They provide an efficient bi-level procedure with a tailored gradient estimator, prove convergence under mild assumptions, and show consistent gains across multiple vision/language benchmarks, with improved sample-efficiency.

**Strengths:**

Thank you for your inspiring work. The main ideas were sound, and the paper delivered them very well. Below are some of the strengths I wish to underline.

- The paper has clear motivations (privacy-aware adaptation) and a compelling idea to address realistic issues in evaluation-based DA.
- The proposed method is sound, and its effectiveness is supported by empirical results.
- The paper is well-written and easy to understand
- Clear Positioning: I appreciate the author's effort in adding Appendix A. Research Position, which concisely captures how the paper relates to other works.

**Weaknesses:**

Below, I have listed some suggestions that would strengthen the paper.


- Statistical Stability: In Sec 4.1. The authors claim that the results of the 'best performance from 10 ind. runs' were reported. I believe that for a fair comparison, the mean performance and its standard error/deviation across #N runs should be reported. Could you please report them?
- The theoretical analysis is sound. However, it relies on strong assumptions (strong convexity and a bounded Hessian), which usually do not hold for most modern deep networks. Could you elaborate on this assumption? Alternatively, the authors could show experiments on small, linear models or provide surrogate approximations.
- Baseline comparisons with DG: I also looked at Appendix E. (comparisons on DG and sDG). However, since DG and sDG do not have target domain access, the comparisons in Tab. 11 & 12 are of less significance (as DA has target domain access). For a fair comparison, the DG/sDG methods should be evaluated with a setting that has a matching feedback budget.
- Hyperparameter Tuning: How were the hyperparameters chosen? In Line 336, you mentioned following the setting in Liu et al., 2018a, but the authors have not shown how changes in $𝜉, γ, ϵ, η, δ$ affect overall performance. Please see Questions.

**Questions:**

Please refer to the weaknesses for additional questions.

- Step Sizes Assumption: The proof assumes diminishing step sizes; however, and correct me if I'm wrong, but experiments appear to use fixed param lr 𝜉 and perturbation lr. Could you explain if these experimental setting aligns with the theory (diminishing schedule).

- Scability & Architectural Adjustments: Do performance gains and query efficiency persist on larger backbones (e.g., larger ResNet variants) or in different architectures (e.g., vision transformers) -- this is of low priority

- Hyperparameters: We want to see how the changes in hyperparameters affect the performance (e.g., Target holdout acc., support-holdout gap, and query efficiency).

---

### Official Review · Reviewer_Ed7R · 2025-11-01

**Soundness:** 2
**Presentation:** 3
**Contribution:** 2
**Rating:** 2
**Confidence:** 5

**Summary:**

This paper introduces Domain Bridging (DB), a novel method for adapting pre-trained models to proprietary target domains without direct data access. The core idea is to learn sample-wise perturbations on the accessible source data, guided solely by performance feedback (e.g., accuracy) from the unobserved target domain. This process steers the model's feature representations to better align with the target domain. The authors propose an Efficient Domain Bridging (EDB) algorithm that addresses the limitations of direct Zeroth-Order optimization by using a more reliable gradient estimation within a bi-level optimization framework. Experiments on image and text classification tasks show that EDB achieves state-of-the-art performance, improving accuracy by approximately 4% over existing baselines.

**Strengths:**

1. This paper presents a novel concept of domain bridging via source data perturbation. This approach effectively narrows the representation gap between source and target domains without violating data privacy, offering a fresh perspective in evaluation-based model adaptation.

2. The method demonstrates consistent and significant improvements across multiple datasets (Office-31, Office-Home, PACS, VLCS, Amazon Review) and modalities (image, text), validating its robustness and generalizability. It also shows faster convergence and better query efficiency compared to baseline methods.

**Weaknesses:**

1. The EDB method depends on a zeroth-order estimator for gradients, which can be noisy and less precise than true gradients. The performance gap between EDB and its variant with exact gradients (EDB) indicates that approximation errors limit the method's full potential.

2. While EDB converges faster than baselines, the process of learning sample-wise perturbations for the entire source dataset within a bi-level optimization framework is inherently complex and could lead to higher computational costs per iteration, especially with large-scale source data.

3. The adaptation process is highly reliant on the performance feedback from the target domain. The paper shows that performance degrades with noisy feedback, suggesting the method might be vulnerable to imperfect or adversarial feedback in real-world scenarios.

4.  The robustness analysis against noisy feedback, while valuable, uses simulated Gaussian noise added to the performance metric. The paper would be more convincing if it tested the method against more realistic noise types, such as label noise in the target domain's support set or non-IID noise distributions that might occur in real-world data.

5. This paper lacks the analysis of perturbation interpretability. The paper correctly notes that the learned perturbations do not visually resemble the target data, which is a privacy feature. However, a deeper analysis of what these perturbations represent or how they correlate with specific domain shift characteristics (e.g., style, texture) would provide valuable insights into the mechanistic interpretation of the "bridging" process, moving beyond quantitative distance metrics like MED.

6.The experiments are conducted on standard academic benchmarks (e.g., Office-31, Office-Home). Although the paper mentions that perturbations can be computed in parallel, it does not fully demonstrate the method's scalability and computational efficiency on larger, more complex datasets (e.g., DomainNet, VisDA 2017) or with larger model architectures (ViT). A more thorough scalability analysis would strengthen the claim for practical deployment.

**Questions:**

See Weaknesses.

---

### Official Review · Reviewer_RAvJ · 2025-11-01

**Soundness:** 3
**Presentation:** 3
**Contribution:** 2
**Rating:** 4
**Confidence:** 4

**Summary:**

The paper proposes a domain bridging framework for adaptation across domains with distributional shifts, where target domain data are proprietary and cannot be directly accessed, using only performance feedback from the target side. This paper proposes an Efficient Domain Bridging (EDB) algorithm that solves a bi-level optimization problem, which (1) finds the best sample-wise perturbation on the source data by optimizing the target domain owner feedback, using an approximated gradient method; and (2) fine tune the model parameters by minimizing the loss on the perturbed source data.

**Strengths:**

1. The paper proposes a domain bridging framework for model adaptation with proprietary target data.

2. The method is evaluated across diverse benchmarks (both image and text).

3. The presentation is generally clear.

**Weaknesses:**

1. **Presentation and notation clarity.**
    Several parts of the paper, particularly Section 2.2, lack sufficient clarity in presentation. The description of Retraining with Source Data Valuation (RSDV) is ambiguous. For example, it is unclear what $\phi_{\pi^t[i]}$ represents, and whether $\phi_{\pi^t[i]}$ and $\phi_{\pi^{t-1}[i]}$ correspond to the same data point. Similarly, the definitions of $V(\theta^t_{i})$ and $V(\theta^t_{i-1})$ are not explicit? Furthermore, the paper should clarify whether the reweighted loss indeed uses $\phi$ values as sample weights. In addition, the notation $\hat{\ell}$ introduced in line 250 does not appear in Eqs. (5)–(6), which disrupts the consistency of notation.

2. **Conceptual positioning and insufficient comparison.**
    The proposed framework appears conceptually closer to zeroth-order (ZO) fine-tuning and adversarial or robustness-oriented perturbation methods than to conventional domain adaptation. The idea of “domain bridging” largely combines these existing techniques and applies them to the scenario where target domain data are proprietary and inaccessible. Because of this hybrid nature, the current comparisons, which focused mainly on ZO and RSDV baselines, are not sufficient to convincingly demonstrate the uniqueness or superiority of the proposed method.

3. **Lack of hyperparameter sensitivity analysis.**
    The paper does not report how sensitive the results are to hyperparameters in Algorithm 1.

**Questions:**

See above.

---

### Note · Authors · 2025-11-17

I have read and agree with the venue's withdrawal policy on behalf of myself and my co-authors.